# Electrophoretic Deposition of Calcium Phosphates on Carbon–Carbon Composite Implants: Morphology, Phase/Chemical Composition and Biological Reactions

**DOI:** 10.3390/ijms25063375

**Published:** 2024-03-16

**Authors:** Andrei S. Skriabin, Petr A. Tsygankov, Vladimir R. Vesnin, Alexey V. Shakurov, Elizaveta S. Skriabina, Irina K. Sviridova, Natalia S. Sergeeva, Valentina A. Kirsanova, Suraya A. Akhmedova, Victoria V. Zherdeva, Yulia S. Lukina, Leonid L. Bionyshev-Abramov

**Affiliations:** 1Department of Power Engineering, Bauman Moscow State Technical University, Moscow 105005, Russia; vesnin.volodya@gmail.com (V.R.V.); shakurovalexey@gmail.com (A.V.S.); elzabra@yandex.ru (E.S.S.); lukina_rctu@mail.ru (Y.S.L.); 2School of Physics, Industrial University of Santander, Bucaramanga 680002, Colombia; piotrtsy@mail.ru; 3P.A. Herzen Moscow Research Oncology Institute, Branch of FSBI “National Medical Research Radiological Centre”, Ministry of Health of the Russian Federation, Moscow 125284, Russia; i.k.sviridova@yandex.ru (I.K.S.); prognoz.01@mail.ru (N.S.S.); kirik-57@mail.ru (V.A.K.); tagieva58@mail.ru (S.A.A.); 4Bach Institute of Biochemistry, Research Center of Biotechnology of the Russian Academy of Sciences, Moscow 119071, Russia; zherdeva.victoria@gmail.com; 5Priorov Central Institute for Trauma and Orthopedics, Moscow 127299, Russia; sity-x@bk.ru

**Keywords:** carbon–carbon medical composites, hydroxyapatite coatings, electrophoretic deposition, in vitro and in vivo tests, radiology studies

## Abstract

Despite a long period of application of metal implants, carbon–carbon medical composites are also widely used for bone defect prosthesis in surgery, dentistry, and oncology. Such implants might demonstrate excellent mechanical properties, but their biocompatibility and integration efficiency into the host should be improved. As a method of enhancing, the electrophoretic deposition of fine-dispersed hydroxyapatite (HAp) on porous carbon substrates might be recommended. With electron microscopy, energy dispersion X-ray and Raman spectroscopy, and X-ray diffraction, we found that the deposition and subsequent heat post-treatment (up to the temperature of 400 °C for 1 h) did not lead to any significant phase and chemical transformations of raw non-stoichometric HAp. The Ca/P ratio was ≈1.51 in the coatings. Their non-toxicity, cyto- and biocompatibility were confirmed by in vitro and in vivo studies and no adverse reactions and side effects had been detected in the test. The proposed coating and subsequent heat treatment procedures provided improved biological responses in terms of resorption and biocompatibility had been confirmed by histological, magnetic resonance and X-ray tomographic ex vivo studies on the resected implant-containing biopsy samples from the BDF1 mouse model. The obtained results are expected to be useful for modern medical material science and clinical applications.

## 1. Introduction

Nowadays, despite a long and successful application of metals (such as titanium and its commercial alloys, magnesium and Co-based alloys, stainless steels, etc. [1,2]) for prosthesis, carbon composite implants [3,4] for bone defect regeneration is under close study due to some carbon advantages as chemical inert, cellular non-toxicity, non-specific foreign body reaction and mechanical properties close to mammal bones [4]. Moreover, some manufacturing features allow for the preparation of a porous surface and inner structures, which can stimulate the required cellular response and accelerate the implant integration into the body [5].

Carbon allotropes such as pyrolitic carbon (PyC), nanotubes (CNT), diamond like carbon (DLC) coatings as well as bulk carbon frameworks could demonstrate bioinert biological behavior in the host body [6,7]. The PyC-based coatings were studied earlier as a thromboresistant barrier on the artificial heart vessels [1]. Wear-resistant and high-adhesive DLC systems were used for heart valves and stents, dental implants, and other orthopaedic supplies [8]. Bulk carbon composites were applied as units for artificial body parts including reconstructive surgery in oncology after tumor harvesting or chemotherapy and metastasis treatment [4] or other applications such as spinal surgery [9] or a drug delivery system [10].

There are a number of approaches to enhance the biocompatibility of framework or scaffold implants, which can include an additive of different organic and non-organic substances in different forms (fibers, coatings, nanoparticles) [11]. In one approach, the carbon fiber composites were reinforced with polyetheretherketone (PEEK) polymer for the manufacturing of orthopaedic implants with mechanical properties close to natural bones [12]. The prospective carbon implants with PEEK reinforcing demonstrated one million fatigue cycles without failure when the wear test allowed them to reach a lower debris volume in comparison to commercial non-reinforced carbon implants. As a model of dental implants [13], the short carbon fiber composites were filled and reinforced with PEEK, which allowed 60% enhancement on the stress distribution through dental implants to the jaw bone.

As an alternative, another prospective approach for stimulation of the implant integration is associated with deposition of different organic [14,15] or non-organic substances [16] on the composite surfaces. Such coatings might stimulate the required cellular response and biological reactions [17] and prevent implant failure. The enhancing of bioactivity of the carbon composites reinforced with PEEK has been studied earlier [14] when the amino groups have been prepared with plasma-enhanced chemical vapor deposition. Surface characterization and a number of in vitro tests with the MG-63 cell culture proved the enhanced water wettability, accelerated cell proliferation, their spreading and adhesion. As a composite matrix [18], non-organic calcium phosphate minerals (such as hydroxyapatite HAp or other bioceramics [19,20]) are a prospective for enhancing biomechanical properties due to their ability to improve the bond strength and eliminate high stresses. Moreover, deposition of crystalline Ca-P coatings can enhance an X-ray radiopacity of amorphous carbon-based grafts that can facilitate surgical procedures during treatment. Based on a complex of electrochemical deposition and chemical treatment procedures [21], carbon fibers were covered with the nano-structured HAp and characterized by scanning electron microscopy (SEM), X-ray photoemission, infrared (IR) and Raman spectroscopies and other techniques. As found, the chemical treatment led to the formation of the O-contained groups, which attracted the Ca^2+^ ions in the electrolyte and stimulated the forming of the chemical bond and HAp nucleation in the form of crystals with an average size from 60 to 210 nm. Different powder spraying procedures were discussed [22] for Ca-P coating deposition on a number of different substrates (including the carbon composites [23]). As shown in [23], atmosphere plasma spraying (DC plasma spraying equipment, 30 kW, 600 A) of raw HAp powders on C-C composites with the next heat post-treatment (at 700 °C for 2 h) allowed for the preparation of high-crystalline coatings with an average shear strength of 7.15 MPa. Similar data on the plasma-sprayed Ca-P layers were obtained with nano- and microsize raw HAp, but some features of the coating structure were in addition highlighted by X-ray diffraction (XRD), energy dispersion X-ray spectroscopy (EDX) and thermal analysis [24]. As demonstrated for the raw nano-HAp, the coating composition was characterized by a presence of crystalline bioactive HAp and cytotoxic CaO [25] unlike micro-HAp particles, whose deposition led to detection of HAp and a small amount of tricalcium phosphate TCP. The biological activity of the nanometer HAp coating was manifested as demonstrated with immersion in simulated body fluid (SBF). As an alternative, detonation spraying was tested [26,27] for deposition of the coatings with the thickness of ≈80–100 μm and the Ca/P ratio of ≈1.67. Such procedures have been considered earlier as a prospective technique to prepare the bioactive calcium phosphate layers also on metal substrates due to low porosity, minimal thermal degradation and enhanced biological reactions [28,29].

High-speed powder spraying (detonation and plasma spraying, etc.) is characterized by a surface deposition of Ca-P powders on the carbon composites without a significant penetration of the particles into internal pores. Despite a high adhesive strength and crystallinity of the thermal sprayed coatings, electrophoretic deposition [30,31] of fine HAp powders may be of some interest in the preparation of Ca-P films on developed and electro conductive surfaces such as porous C-C composites. But there are data gaps in the complex of biological reactions between tissues or cells and the covered composite implants.

So, the aim of the presented study was to establish the relationships between composition, morphology of the electrophoretic deposited coatings and biological reactions.

The routine cellular tests for cytotoxicity, cytocompatibility and osteogenic differentiation allowed us to obtain a quantitative and statistically significant data on the cells’ interaction with the covered and original (uncovered or as-prepared) carbon implants followed by their implantation into mice with subsequent resection of biological samples for further ex vivo morphological study combined with radiological methods.

## 2. Results

### 2.1. Morphology and Properties of as-Prepared C-C Composites and HAp Powder

SEM images of the agglomerates from the raw nano-HAp powder (with an average size of ≤60 nm) are presented in Figure 1a,b. Its Raman spectrum is shown in Figure 2a. In our previous study [28], the Ca/P ratio has been ≈1.45–1.50 for this commercial phosphate. With Raman spectroscopy data, the recorded weak peaks at ≈1068 cm^−1^ and ≈530 cm^−1^ could be explained by the ν_1_[CO_3_]^2−^ and ν_4_[HPO_4_]^2−^ vibrating modes [32,33]. The other recorded modes (at ≈424 cm^−1^, ≈581 cm^−1^, ≈955 cm^−1^ and ≈1039 cm^−1^) were typical for different vibrations of the [PO_4_]^3−^ ions.

The photos of the original C-C composite substrates (see Figure 1c), their cross-section optical and SEM images (Figure 1d–g) are presented below. As found, the photos demonstrated a presence of large inner circle-like pores with an average size of ≈500 μm (see Figure 1d) and smaller holes (see Figure 1e). Generally, the studied composites had a framework structure with a number of cavities, which could be a depot for extra-fine HAp powder after deposition.

Amorphous carbon binding had a layered structure. Due to high porosity and the layered structures, the visualized composite structure could stimulate adsorption of water vapors that were partially confirmed by energy dispersion X-ray spectroscopy (EDX): the element composition demonstrated a presence of a large oxygen concentration (up to ≈10.81 wt.%). Hydrogen atoms were not detected because of a low EDX sensibility for these atoms. The used machining processing with a diamond tool did not lead to the detection of impurities.

Presented in Figure 2b, the Raman spectrum of the C-C composite demonstrated a presence of the in-plane C-C bond stretching G (≈1600 cm^−1^) induced by the lattice defects D (≈1360 cm^−1^) modes that corresponded to amorphous carbon.

### 2.2. Morphology and Properties of Ca-P-Coated C-C Composites

The obtained SEM images evidenced a presence of deposited HAp at different scales as presented in Figure 3. The C fibers (with the average diameters of ≈6–8 μm) were dotted with Ca-P fragments, and the gaps between binding carbon layers also contained individual HAp (see Figure 3a–d). We should note HAp covered the composites on all sides including their end sides (Figure 3e–h) and lateral surfaces (Figure 3i–l). Using backscattering electron images in Figure 3m–p, a SEM image of the well-contrasted HAp coatings demonstrated a continues spreading on the surface with an un-uniform thickness. Individual cracks in the coating were visualized as in Figure 3l.

At small scales, the film discontinues were discovered, i.e., the HAp layer was formed by a merging of low-dimension HAp fragments during the deposition (Figure 3p). As found, the used electrophoretic deposition led to a pervasion of HAp fine particles into the small and large pores and fixation on carbon walls. As shown in Figure 3q,r, the EDX spectrum of the coating evidenced a presence of such typical for calcium phosphates elements as Ca, P and O. The emission of carbon lines was caused by a substrate influence. The Ca/P ratio (in atomic %) was estimated with the EDX data on the Ca and P concentration and equaled to ≈1.51 close to the raw HAp. No contaminants were detected due to the low sensibility at these measurements.

The Raman spectrum (see Figure 2c) of the deposited HAp coatings demonstrated all the main vibration modes typical for the raw HAp (see above). No broadening of the major ν_1_[PO_4_]^3−^ peak at ≈955 cm^−1^ was detected that demonstrated a negligible phase/chemical transformation in the HAp layer during preparation and further heat post-processing. For these studied coatings, a poor indication of the ν_1_[CO_3_]^2−^ and ν_4_[HPO_4_]^2−^ modes was caused by the laser probe light scattering on the complex framework surfaces. The above data on the phase composition were confirmed by the XRD data (see Figure A1 in Appendix A), which corresponded to a composition of crystalline hexagonal HAp (PDF No. 9-432) and amorphous phosphates.

### 2.3. Results of In Vitro Tests

#### 2.3.1. Cytotoxicity

The results of the cytotoxicity tests are presented in Table 1. The optical density (OD) values are presented as the means ± standard deviations. The data on the population of viable cells (PVC) and a toxicity index (TI) showed an absence (TI < 30%) of acute toxicity for all studied samples (complete grow medium (CGM) as control, as-prepared and Hap-covered composites as group 1 and group 2, respectively). After 72 h of the MG-63 cells (human bone osteosarcoma) growth, the TI values were 0%, 0% and 15.1%, respectively.

The cytotoxicity data were confirmed by the Live/Dead test for 24 and 72 h of growth, and the appropriate photos are presented in Appendix A. Figure A2 demonstrates the results of phase contrast and fluorescence microscopy studies of the MG-63 cells for the negative control and two test groups, which evidenced a high viability (green fluorescence) and a minor dead population (red fluorescence) at the indicated periods. Moreover, a significant increasing rate (hour 72 vs. hour 24) of the cells for all samples was found with no significant differences between the groups.

#### 2.3.2. Cytocompatibility

During co-cultivation of the C-C composites with the test cells during 14 days, a stable cytocompatibility for all groups was found. We detected a statistically significant difference (*p* * < 0.05) in the PVC values between group 2 and the control for 1 (84.9%) and 3 (70.9%) days. In terms of the optical densities, the OD values were 0.152 ± 0.01 (0.179 ± 0.01 for the control) at day 1 and 0.439 ± 0.01 (0.619 ± 0.03 for the control) at day 3. The statistically significant difference of the PVC and OD values between the groups revealed the cultivation duration up to day 3.

Afterwards, the situation changed. The cell population was increasing progressively and was not statistically differing from the control in the OD and PVC values for 7 days and later as well as between the groups. The final (at day 14) PVC and OD values reached 95.6%, 3.307 ± 0.04 (group 1) and 92.6%, 3.204 ± 0.07 (group 2), respectively. The appropriate data on the PVC and OD increasing rate are presented in Figure 4a,b. As a demonstration, the population dynamics of the MG-63 cells for the control growth and the group 1 and group 2 samples are also presented in Appendix A (Figure A3).

#### 2.3.3. Osteogenic Differentiation

The study of osteogenic differentiation of the cells on the C-C composites was carried out with the MSCs of the donor’s bone marrow, which corresponded to the immunophenotype with CD90 + CD105 + CD73 + CD34-CD45- expression.

In the case of the control, the manifest expression of RUNX2 and ALPL genes was detected at the FD levels of 2.5 ± 0.2 and 81.6 ± 20.3, respectively. In the case of the experimental groups, all samples strictly maintained the differentiation, and the maximum expressions of RUNX2 and ALPL genes were found as 3.0 ± 0.3 for RUNX2 and 163.1 ± 48.5 for ALPL for the group 1. The group 2 samples had slightly lower values (2.6 ± 0.2 for RUNX2 and 112.7 ± 30.5 for ALPL) of the expressions as presented in Figure 4c. No statistically significant results were found during this test.

### 2.4. In Vivo Tests

#### 2.4.1. Histological Patterns

At 6 and 12 weeks after the implantation, the animals were sacrificed for the morphometric and histological analysis of the samples. 

A fragment of a non-uniform earlier stage (6 weeks) connective tissue capsule for group 1 is presented in Figure 5. There are no signs of inflammation around the implants. A thin region of the capsule was formed by a fairly dense fibrous connective tissue containing fibroblastic differing and filled with individual capillaries (see Figure 5a–d). A loose region was formed by rather haphazardly located collagen fibers and capillaries with different diameters. A more manifest vascularization was detected around the carbon fragments as shown in Figure 5c,d. These facts indirectly testify to a weak tissue irritation through contact with the C-C composite surface.

After 12 weeks, a region of the thin mature and paucicellular capsule was visualized (see Figure 5e,f) as the tissue with a poor vascularization around the tissue fibers. The space of individual implant pores was filled with active fibroblasts forming an extracellular matrix or was paved with the paucicellular connective tissue as demonstrated in Figure 5g,h.

In the case of the group 2 implants, a fragment of thin paucicellular capsule with individual vessels in adipose tissue was visualized for 6 weeks and presented in Figure 6a,b. Two fragments of the capsules around such samples for 12 weeks are shown in Figure 6c–f: the first fragment is represented by a thin and properly formed paucicellular connective tissue (see Figure 5d and Figure 6c). The second fragment was extracted from the inner pore surface and contained loose and correctly arranged connective tissue fibers with the material (see Figure 6e,f). As for the above cases, there are no inflammation and bacterial infections in the surrounding tissues. As found with histological studies, some cell features of the implant integration into the surrounding tissues as well as an absence of inflammation, giant cells and other reactive changes testifies to the biocompatibility of all the studied C-C composites. Obtained with mechanical destruction of the “implant/tissue” complex, the final assessment on the HAp role for implant integration must be proven with a non-destructive procedure (e.g., MRI and CT).

#### 2.4.2. Radiological Studies

Harvested at 12 weeks, the composite implants (group 1 and group 2) were studied using MRI and CT. The appropriate MR and CT patterns are shown in Figure 7. These samples looked in appearance as the implants surrounded by the tissue capsule adherent to the skin.

MR T2w images (see Figure 7a–d) demonstrated a strong hypointense signal for the implant, and moderate hyperintense foci were detected on all sides. Morphologically, it looked like a bursa of fibrous-focal nature according to the histology results (see Figure 5 and Figure 6). An adjoined skin fragment was also visualized as a hypointense region. T1w images (see Figure 7e,g) for as-prepared C-C implants demonstrated some signs of the degradation. The capsule and skin were easily differed from the implant due to the high contrast of the appropriate region of interests (ROI) (see Figure 7e–h). The adjoined skin was visualized by a strong hyperintense signal due to the short relaxation time in T1-weighted sequences. Partial and poor resorption of the implant-like implant shape change and hypointensity level change were found as well as individual resorption foci. A more intensively extent resorption process (Figure 7f,h) was detected for the group 2 samples in comparison to the group 1 samples (Figure 7e,g).

The weak hyperintense signal from the skin was captured with T1w fat-suppressed imaging (Figure 7i,j). At the same time, the MR signal from the connective tissue capsule was differed for the group 2 and group 1 samples captured using the same T1w fat-suppressed mode. Figure 7i,j represents the fat-suppressed images after an application of a transverse impulse of 100 ms. Thus, the prevalence of the fibrous capsule signal over the suppressed adipose tissue signal was demonstrated, that allowed us to segment and estimate the MR signal from the fibrosis capsule at the scaffold resorption.

According to Figure 7b,d, the morphology of the tissue around the implanted HAp-covered scaffold demonstrated the looser connective tissue capsule. With fat-suppressed T1w imaging, the SNR reducing of MR intensity for the fibrosis capsule was strongly revealed for the HAp-covered implant compared to the as-prepared samples. In this case, the SNR values for the capsule had a difference of 1.27 times between the groups (*p* * < 0.05), as shown in Figure 7k, thus confirming the more pronounced resorption of the group 2 samples compared to the group 1 samples at the 12-th week.

With the CT data (see Figure 7l,m) on the tissue densities, the group 1 samples were visualized in the tissue capsule with a detection of relatively high radiographic density regions, but its shape was blurred due to the influence of surrounding soft tissues. Generally, such high densities (≈30–40 HU) were observed inside the composites. In the case of the group 2 implants, the CT pattern was somewhat different, but the implant profile was also formed by the contours of cellular regions with a high density. The profile of the high-density region was saved, and the tissues around the HAp-coated implants had a lower density value compared with media in the cellules (≈10–20 HU). But for some locations at the “implant/tissue” interface, the individual local consolidations (with the densities of ≈40–50 HU) were found due to the coating migration from the surface or the sample resorption.

## 3. Discussion

Nowadays, Ca-P bioceramic coatings are deposited to enhance implant integration into bone tissue but understanding and describing the relevant biological processes are difficult. The presented results could be considered as a necessary preliminary stage of biotesting of the HAp-covered C-C composites with in vitro and subcutaneous implantation procedures [34,35].

As found earlier in [24] for micron and nanoscale HAp powder deposited with plasma spraying on the C-C composites, SBF soaking confirmed good bioactivity for 14 days of the immersion duration indicating the formation of the bone-like matrix. As an alternative method to the HAp coating study [36], the in vitro bioactivity tests demonstrated the enhanced cell proliferation and osteogenic markers expression for the HAp coating prepared by electro-deposition on the C-C composites. The biocompatibility of the HAp-coated C-C materials was confirmed in the MTT-test, demonstrating a high level of proliferation [36]. As another deposition strategy [37], the HAp coatings (converted from deposited monetite CaHPO_4_) on the carbon composites also demonstrated good activities of the ALPL expression with a significant level of *p* * < 0.05. Fluorescent and electron microscopy allowed them to study the mature degree of differentiation of osteoblasts. In the case of the carbon foam/HAp coating [38], the in vitro tests with the MG-63 cells demonstrated better activity and cell adhesion compared with the uncovered composites as found with SEM. The fluorescent microscopy and the MTT-test confirmed that both cell adhesion and cell proliferation were better for the covered substrates. Here we should note that the RUNX2 gene expression is very important for the comprehension of osteobalstiogenesis [39], but such data have not been sufficiently represented yet. Moreover, in vivo studies are absolutely necessary for an expansion of the implant integration picture into the host.

As found with XRD, SEM, EDX and Raman spectroscopy in our study, all of the detected compounds (non-stoichiometric HAp coating and amorphous carbon framework) in the samples were biocompatible. The phase composition of the coatings was mainly formed by a presence of the [PO_4_]^3−^ groups, which are typical of Ca-P-based materials, and minor concentrations of the [CO_3_]^2−^ and [HPO_4_]^2−^ ions. Only hexagonal HAp and amorphous phosphates were found. The detected compounds are absolutely typical of bioactive ceramics [19,20]. The electrophoresis deposition procedure did not lead to a formation of any cytotoxic components (as in the case of plasma spray [40], which thermally decompose raw HAp into CaO, amorphous Ca-P, tetracalcium phosphate, etc. [41]). As is well-known [30,42], overheating causes partial HAp crystallization (above 400 °C) and the rearrangement (at 780 °C) of the substitutive [CO_3_]^2−^ and [HPO_4_]^2−^ ions with extraction of gaseous CO_2_ and H_2_O and a shifting of the Ca/P ratio to ≈1.67. So, the used post-treatment (a heating up to 400 °C for 1 h) did not also lead to significant chemical transformations of the deposited HAp or the Ca/P ratio changing, but it was relatively effective in removing cytotoxic isopropyl alcohol from the inner pores. The porous C-C composite contributed to the penetration of fine-dispersed HAp from the colloidal electrolyte and its accumulation. Generally, the phase composition of our prepared coated samples was typical for the previous published data [35,36].

In the present study, we presented some results of biocompatibility and toxicity tests for the prepared coatings. As found (see Table 1 and Figure 4) for group 2, the PVC and TI values were statistically significantly (*p* * < 0.05) poorer than the control and the group 1 samples during the earlier stage of the cell growth (up to day 3 of the cultivating). In the case of the more prolonged cultivation duration (starting from day 7), we detected no significant difference between the control and the groups as well as between group 1 and group 2. In our opinion, a probable cause was the presence of a trace amount of surface impurities in the samples. The prolonged cultivation with the regular CGM replacement could decrease their concentrations and the PVC values were statistically the same between the groups and the control starting from day 7 to day 14. So, non-toxicity (IT ≤ 30%) and cytocompatibility (PVC ≥ 70%) of all studied samples were confirmed by in vitro tests. The group 1 and group 2 samples maintained osteogenic differentiation (the target genes of RUNX2 and ALPL) of the donor’s BM MSCs. Generally, the obtained results confirmed the previous published data [35,36,37] on the in vitro cytocompatibility potential of the prepared HAp coatings in terms of the ALPL gene expression and cell proliferation. As shown additionally, the pronounced RUNX2 expression was detected for the studied samples too.

The prepared histological patterns evidenced a tight-fitted capsule, a negligible inflammation in the surrounding tissues and an absence of bacterial infections. Formed during the sample preparation, a significant amount of composite fragments were visualized. According to Hench’s classification [43], the group 1 implants (as-prepared composites) demonstrated biocompatible properties because of a presence of high-contrasted collagen fibers in the connective tissue capsule with no traces of inflammation and bacterial lesion (see Figure 5). At the same time for the group 2 surfaces, moderate lymphocyte and macrophage infiltration and a lot of capillaries were observed around the implants. According to the results obtained earlier, a less dense connective tissue capsule and more vascularized surrounding tissue contributed to a better biointegration of the implant [44]. Similar results were obtained in our study.

In addition, histological data were confirmed in an MRI study based on varying degrees of contrast between linearized and loose collagen of the fibrous capsule. The histological patterns allowed us to make a detailed study of the tissue capsule fragments. The obtained radiological data (see Figure 7) carried the information about the density of the fibrosis capsule and degree of collagen linearization.

The contrasting of collagen was demonstrated earlier [45], where the 3T MRI was used to determine linearized collagen associated with prostate tumors. We presumed that in our samples, a high relaxation of water protons in connective tissue (hyperintense signal on fat-suppressed T1-weighted images) occurred as a result of sample formalin fixation. The buffered formalin fixated the tissues by cross-linking the amino groups via a methylene bridge, thereby inactivating proteins and stabilizing them. It was shown earlier that such chemical fixation affected the acquisition of diffusion-weighted images [46]. Hence, the change of a hyperintense signal of a fat-suppressed T1-weighted image according to the discussing above may reflect the order of collagen. As revealed, the fat-suppressed T1-w signal was more hyperintensive for the group 1 scaffold and less for the group 2, thus pointing at a low degree of orderliness. Our macroscopic MRI results were confirmed by histological results as well. Caused by the coating biodegradation, the emergence of resorption foci (with the density decreasing from ≈30–40 HU up to ≈10–20 HU) in the implanted covered structures was also found with CT.

So, the loose and vascularized tissue infiltrated by immune cells is a sign of biocompatibility of the HAp-covered samples (group 2). In contrast, a long-pronounced encapsulation evidenced a less biocompatible degree of the as-prepared composites (group 1). Such an encapsulation may be caused by a reaction due to the implantation into the body of a foreign material non-interacting with the tissues. At the whole-body level, this could continue through scar tissue resorption or a prolongated scarring that is determined by the body’s reactivity [43].

Currently, some features of the implant integration into bone are discussed [47] in terms of surface characteristics, chemical composition, deposited coatings, etc. As is known [48,49], HAp and other Ca-P-based compounds can also enhance soft tissue repair and cell reactions around the implants due to the affinity between HAp and collagen stimulating collagen formation with suppression of inflammation and failure. Additionally, deposited HAp could change the surface morphology and increase its roughness that could stimulate the cell adhesion to a substrate resulting in biocompatibility enhancement [50]. Additionally, deposited HAp crystals and other Ca-P compounds are metabolized in the body to the Ca and P ions, which can be included into the structure of regenerated bone tissue [51] or collagen fibers [52].

So, the completed electrophoretic deposition of Ca-P layers on porous carbon implants can stimulate the biocompatibility of the C-C composites with no adverse reactions.

## 4. Materials and Methods

### 4.1. C-C Composites and HAp Powder

The commercial C-C composites (NTM+ Ltd., Vsevolozhsk, Russia) [53] were used as substrates in the form of cylinders with a diameter of ≈6 mm and a length of ≈3 mm. Commercial non-stoichiometric calcium-defied HAp (99%) powder (Biteka Ltd., Odintsovo, Russia) had a maximum particle size of 60 nm. In our previous study [28], the carbonate [CO_3_]^2−^ and monohydrogen phosphate [HPO_4_]^2−^ ions were detected in the powder structure and thus the Ca/P ratio was equaled to ≈1.45–1.50.

### 4.2. Deposition Procedure

HAp coatings were obtained with electrophoretic deposition at a DC voltage of 300 V for ≈1 min. The electrolyte included the HAp suspension (≈10 g/L), which was prepared with the ultrasound dispersion at 60 W and 75 °C for 30 min in isopropyl alcohol (99.7%). For elimination of possible cytotoxic precursors, the prepared samples were washed in distilled water and heat-treated under atmosphere at 400 °C for 1 h.

### 4.3. Coating Characterization

The chemical composition of the C-C composite, raw HAp and coatings were studied with an ATR2500 Raman spectrometer (Optosky, Xiamen, China) with a central probing wavelength of 785 ± 0.5 nm and the spectral range of 250–2700 cm^−1^ at a resolution of 4 cm^−1^. The phase composition was investigated with X-ray diffraction XRD (DRON-3M diffractometer, Burevestik Ltd., St.-Petersburg, Russia; CuKα radiation) within the angle range of 2θ = 20−65° and a stepsize of 0.02°. The XRD patterns were interpreted with ICDD PDF-2 database [54].

The morphology and structure of the samples were investigated by scanning electron microscopy (SEM) with a Quattro microscope (Thermo Fisher Scientific, Waltham, MA, USA; electron energy of 5−10 kV, magnification of ×400−3500). The element composition of the coatings was determined with EDX (EDAX Octane Elect plus energy dispersive analysis system).

### 4.4. In Vitro Tests

#### 4.4.1. Sample Sterilization

Before the tests, the samples were dry and heat sterilized at 180° C for 90 min (Binder GbmH, Tuttlingen, Germany).

#### 4.4.2. Cell Cultures

In vitro experiments were carried out on the osteoblast-like cells of human osteosarcoma MG-63 (CRL-1427TM, American Type Cell Collection (ATCC)) and with the donor-derived primary cell culture of human bone marrow (BM) mesenchymal stem cells (MSCs), 3–8 passages. Obtaining an intraoperative sample of biological material from patient A was performed after voluntary informed consent of the patient was obtained and approved by the Independent Ethics Council of the P.A. Herzen Moscow Cancer Research Institute.

The MG-63 cell growth was carried out at an initial seeding density of 4.0 × 10^4^ cell/cm^2^ in the complete growth medium (CGM) based on DMEM medium (PanEko, Moscow, Russia), 10% fetal bovine serum PAA (HyClone Laboratories Ltd., Washington, DC, USA), glutamine (0.58 mg/mL, PanEko, Moscow, Russia), HEPES solution (20.0 μM, PanEko, Moscow, Russia) and gentamycin (50 μg/mL, Dalkhimpharm, Khabarovsk, Russia).

The MSCs were maintained at a sowing density of 1.0 × 10^3^ cell/cm^2^ in the CGM based on DMEM/F12 medium (PanEko, Moscow, Russia). The MSCs were certified by immunophenotyping with the monoclonal antibodies to CD105, CD90, CD73, CD45, CD34 markers (Miltenyi Biotec, Bergisch Gladbach, Germany) and had the following phenotype: CD105+, CD90+, CD73+, CD45-, CD34-.

All manipulations with the cell cultures were performed under the standard aseptic conditions in humid air at 37 °C and 5% CO_2_ (CO_2_ incubator; Sanyo, Osaka, Japan). The medium was substituted twice a week. Cell cultures in the preconfluent monolayer were used for experiments.

#### 4.4.3. Cytotoxicity Tests

Cytotoxicity was estimated in accordance with ISO 10993.5-99 [55] by indirect contact of the extracts and the cells and direct contact by the cell seeding on the samples.

##### Indirect Contact Test

To prepare the extraction solution, 0.1 g of sterile sample was placed in 1.2 mL of CGM incubated for 24 h at 37 °C with constant stirring at an orbital shaker (Elmi, Riga, Latvia). The MG-63 cells with a seeding density of 20.0 × 10^3^ cells/cm^2^ (96-well plates; Corning, Summerville, MA, USA) were cultured until a subconfluent monolayer was reached with the following complete replacement of CGM with sample extracts. After 24 and 72 h, the viability of the cell culture was analyzed using the MTT assay.

As negative and positive controls, CGM and 50% dimethyl sulfoxide (PanEko, Moscow, Russia) were added to the MG-63 cell cultures, respectively. At least five extraction probes were studied for each sample and control.

##### Direct Contact Test

The sterilized composites were placed into the wells of 96-well plates for cell growth (Corning, Summerville, MA, USA) (each type of sample in triplet for each period) and the MG-63 suspension (with a seeding density of 30.0 × 10^3^ cells/cm^2^) was added. Cell growth periods were the same as for the indirect test. As a control, the cells on the polystyrene were used. The cell spreading along the surface was control visually with a stereo zoom microscope (Olympus, Tokyo, Japan) and a Ti Eclipse microscope (Nikon, Tokyo, Japan). Next, the viability of the MG-63 cell culture was analyzed using the MTT assay.

#### 4.4.4. Cytocompatibility Test

In vitro cytocompatibility was evaluated by a direct contact of the test cells and the studied composites. The sterilized samples were placed into the wells of the 24-well plates (Corning, Summerville, MA, USA) in the presence of the MG-63 cell suspension with a seeding density of 20.0 × 10^3^ cells/cm^2^. With a polyester control, the samples were cultivated for 1, 3, 7, 10 and 14 days with a regular (twice per a week) change of suspension.

#### 4.4.5. Assessment of Cell Culture Viability

##### Assessment of Cell Culture Viability Using the Quantitative MTT Assay

The MTT assay is based on the ability of live cell dehydrogenases to reduce 3-(-4.5-dimethylthiazolyl-2)-2,5-diphenyltetrazolium bromide (MTT, Sigma-Aldrich, St. Louis, MO, USA) into water-insoluble blue formazan crystals [56]. To perform the MTT test, an MTT solution at a concentration of 5.0 mg/mL was added to the cell culture medium. After 3 h of incubation (5% CO_2_, 37 °C), the formed formazan was dissolved with isopropyl alcohol, and the remains of lysed cells were precipitated via centrifugation (10 min at 3000 rpm). In the resulting supernatant, the optical density (OD) of the formazan solution was evaluated on a Multiscan FC spectrophotometer (Thermo Scientific, Waltham, MA, USA) at a wavelength of 540 nm.

The calculation of the population of viable cells (PVCs) in relation to the control (in %) was carried out according to
PVC = OD_test_/OD_control_ · 100%.

Here OD is the optical density of the formazan solution. As the control, the non-treated cell culture was incubated under standard conditions.

Further, a toxicity index (TI) was calculated as
TI = 100% − PVC.(1)

When TI ≤ 30% and PVC ≥ 70%, we considered the studied samples as non-toxic and cytocompatible implant prototypes. At the achieved time points, the dynamics of cell spreading was visualized and captured.

##### Assessment of Cell Culture Viability Using a Live/Dead Kit

A semi-quantitative evaluation of cell viability was fulfilled with a Live/Dead tool kit (Molecular Probes, Eugene, OR, USA) in accordance with the manufacturer’s protocol for visualization of the live and dead cells with the direct contact and simultaneous imaging of live (with Calcein-AM) and dead (with Eth-1 dye) cells.

#### 4.4.6. Osteogenic Differentiation of MSCs on Composites Using PCR

The MSC_S_ were cultivated on the culture plastic, original and covered C-C composites. Directed differentiation of BM MSCs into osteoblasts was performed by culturing in the osteogenic medium StemPro^TM^ Osteogenesis Differential Kit (Thermo Fisher Scientific, Waltham, MA, USA) for 2 weeks using the manufacturer’s standard protocol. The differentiation efficiency was assessed on day 14 by measuring the expression of the RUNX2 and ALPL osteoblastic marker genes using real-time PCR.

##### PCR Test

The osteogenic differentiation genes (Runt-related transcription factor 2 RUNX2, alkaline phosphate ALPL) were analyzed on day 14, taking into account the assessment of the housekeeping gene (GAPDH) for the expression normalization. The total RNA pool was isolated from the MSCs culture after the cultivation by phenol-chloroform extraction with a Qiazol lysis buffer (Qiagen, Hilden, Germany) followed by condensation and purification of nucleic acids with an RNeasy Mini Kit columns (Qiagen, Hilden, Germany). RNA was two-step eluted with RNase-free water. A final eluate volume was 40 μL per each sample.

Quantitative analysis in the samples was carried out with a NanoDrop ND-2000 spectrophotometer (Thermo Scientific, Waltham, MA, USA). A commercial MMLV RT toolkit (Eurogen, Moscow, Russia) with an adding of Random (dN)_10_-primers was used for a reverse transcription and obtaining of complimentary DNA. We used a qPCRmix-HS assay with SYBR Green I intercalating dye (Eurogen, Moscow, Russia) for the PCR test. The signal was recorded in real-time with a DTlite detecting amplifier (DNA Technology, Moscow, Russia). The amplification program included a “hot start” (94 °C, 10 min) followed by 50 cycles of template denaturation (94 °C, 20 s), primer annealing (64 °C, 10 s), and amplicon elongation (72 °C, 15 s). All samples were analyzed in triplets.

The expression of the studied genes was analyzed using the ΔΔCt algorithm. With this algorithm, the expression level of the studied genes in both the test samples and control samples was adjusted in relation to the expression of normalizer gene (GAPDH) from the same two samples. As a result, the fold difference (FD) in the expression level of the studied gene in the test and control groups was calculated. The control was non-treated cell culture incubated under the standard conditions. Oligonucleotide sequences of the target and normalized primers (see Table 2) were synthesized by Eurogen (Moscow, Russia), J.S.C.

#### 4.4.7. Statistical Analysis

Statistical analysis of the obtained data was performed with Student’s *t*-test (Microsoft Excel) at a significance level of *p* * ≤ 0.05.

### 4.5. In Vivo Tests

#### 4.5.1. Implantation and Histology Procedures

All in vivo tests were provided with the ethical rules and requirements formulated in the European FELASA Directive 2010/63/EU on the protection of animals used for scientific purposes. The animals were kept in the experimental vivarium at P.A. Hertsen Moscow Oncology Research Institute under the standard light, food and water conditions.

Biocompatibility was tested with the use of the implantation model into the adipose tissue of male BDF1 mice weighted ≈ 18–20 g (Federal Medical-Biological Agency, Andreevka branch, Russia). The sterilized at 180° C for 90 min (Binder GbmH, Tuttlingen, Germany) samples were implanted subcutaneously into caverns on the back under antiseptic conditions and general anesthesia by intra-abdominal injection of 0.1 mL ketamine/relanium mixture (the concentration ratio of 1:1). Two experimental groups were formed for the as-prepared (group 1) and covered C-C composites (group 2) as shown in Table 1. The total animal amount was 4 per a group. After 6 and 12 weeks of cell growth, the mice (two animals per each date) were euthanized with a CO_2_ chamber (Techniplast, Buguggiate, Italy). After animal autopsy, the samples of scar tissues joint with the implants were harvested and fixed in 10% buffered formalin. After the fixation, the capsule edge was dissected by an eye scalpel and then the soft tissues were separated away from the composite by a needle. The capsule volume was filled with a warm agarose solution (Sigma-Aldrich, St. Louis, MO, USA) and standard histological sample preparation was carried out.

Further the capsules were dissolved in an agarose solution, paraffin-embedded; and then histological slides with a thickness of 4 mm were prepared and stained by hematoxylin–eosin. Light microscopy was performed with a Ti Eclipse microscope (Nikon, Tokyo, Japan).

#### 4.5.2. Magnetic Resonance Imaging and X-ray Computer Tomography

After harvesting, the group 1 and group 2 samples were placed into 15 mL tubes and filled with 10% neutral buffered formalin. 1T ^1^H M3™ Compact High-Performance Magnetic Resonance Imaging (MRI) System (Shoham, Israel) with the 50 × 150 mm inner bore was used for pattern capturing. A sample-contained tube was placed into the integrated handling system with the body RF coil (with a diameter of 38 mm and a length of 50 mm) used for image acquiring. MR images were acquired at the 20–40 min intervals using T2w FSE (T2-weighted fast spin echo) (TR/TE 4000/38.3, FOV 40 × 40 mm, 120 × 180, ETL 5, NEX 8–15 sequences) and T1w GRE (T1-weighted 3D gradient echo) (TR/TE = 60/2.4, FA 30, NEX 20, FOV 40 × 40 mm, 120 × 180, NEX 8–15 sequences).

The TI = 100 ms inversion pre-pulse was applied for suppression of fat imaging. The obtained images were analyzed by a selection of region of interests (ROI) using 3–4 scans with a thickness of 1 mm using Image Fiji open software (National Institute of Health, Stapleton, NY, USA). The differences in values of SNR changes were considered statistically significant (*p* * < 0.05) according to the one-tailed non-parametric Mann–Whitney-Wilcoxon (MWW) *t*-test paired observation, and the data were analyzed using the one-tailed non-parametric (unpaired) MWW *t*-test, *n* = 6–12 data points per a group. The multiple groups were compared using the ordinary one-way ANOVA test (GraphPad Prism version 9.0.0, GraphPad Software Inc, Boston, MA, USA).

X-ray CT images of the buffered samples were visualized with a SkySkan 1178 scanner (Bruker, Billerica, MA, USA) with the array resolution of 1024 × 1024 pixels and a voxel size of 84 μm. 3D structures of the “tissue/implant” complex were reconstructed and visualized with VSG Avizo software (version 8.0, VSG Software Solutions Ltd., Kartanaka, India). On the Hounsfield units (HU), we used SkySkan CTan software (version 1.18, Bruker Corporation, Billerica, MA, USA) for estimation of the tissue densities around and inside the porous carbon compositions.

## 5. Conclusions

Medical C-C composites are currently used for bone defect repair in trauma healing, oncology prosthetics and maxillofacial surgery. In the present study, we developed the strategy for enhancing of necessary biological reactions in the host by the deposition of HAp coatings. As found with XRD, SEM, EDX and Raman spectroscopy, the electrophoretic deposition with the mild heat post-treatment (at 400 °C for 1 h) led to a formation of the non-stoichometric HAp coating with the Ca/P ratio of ≈1.51 and a presence of the of the major vibrating ν_1_[PO_4_]^3−^ mode in the Raman spectra and the pronounce XRD pattern of hexagonal apatite. The HAp covered the substrate and penetrated into the pores. In vitro tests demonstrated that such a strategy allowed us to prepare the non-toxic and cytocompatible covered samples, which supported the osteogenic differentiation of the MSCs (the target genes of RUNX2 and ALPL). Using the in vivo BDF1 mouse model of subdermal implantation, the obtained histological data demonstrated suppressed inflammation reactions. As found, the prepared coatings caused a formation of a tissue capsule with a moderate lymphomacrophage infiltration, which was looser than for the uncovered composites. The used MRI procedure allowed us to characterize the capsule structure around the implants as poorly ordered. The data on histological and radiological patterns could be considered as signs of a good biocompatibility. The X-ray CT evidenced a high-density region formation due to the coating migration from the carbon surface.

The obtained data are useful as the first step for understanding the perspectives of HAp-covered carbon medicine materials and biological cell/soft tissue reaction around such implants for bone defect healing. The implant integration into bone tissue will be studied in future.

## Figures and Tables

**Figure 1 ijms-25-03375-f001:**
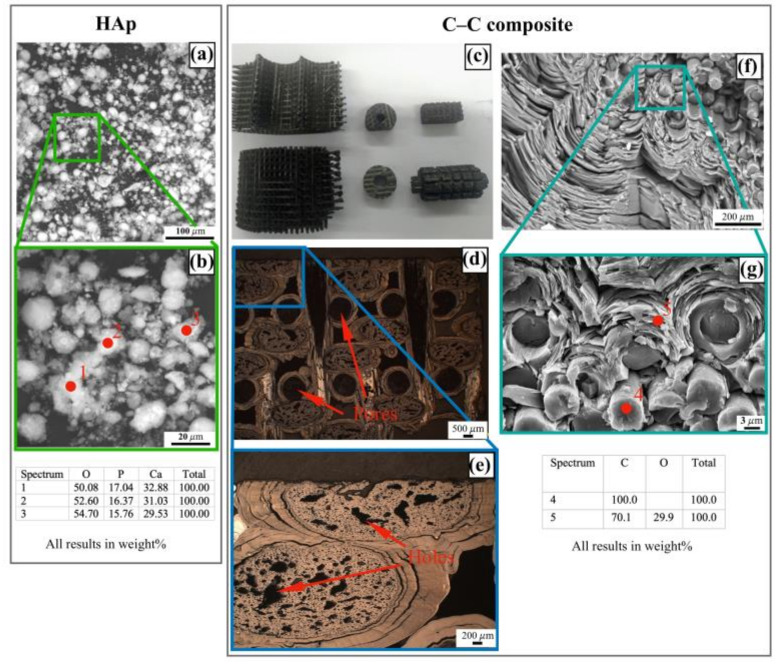
SEM images of raw HAp (**a**,**b**) and photo (**c**) and cross-section images with EDX data (**d**–**g**) of as-prepared C-C composite.

**Figure 2 ijms-25-03375-f002:**
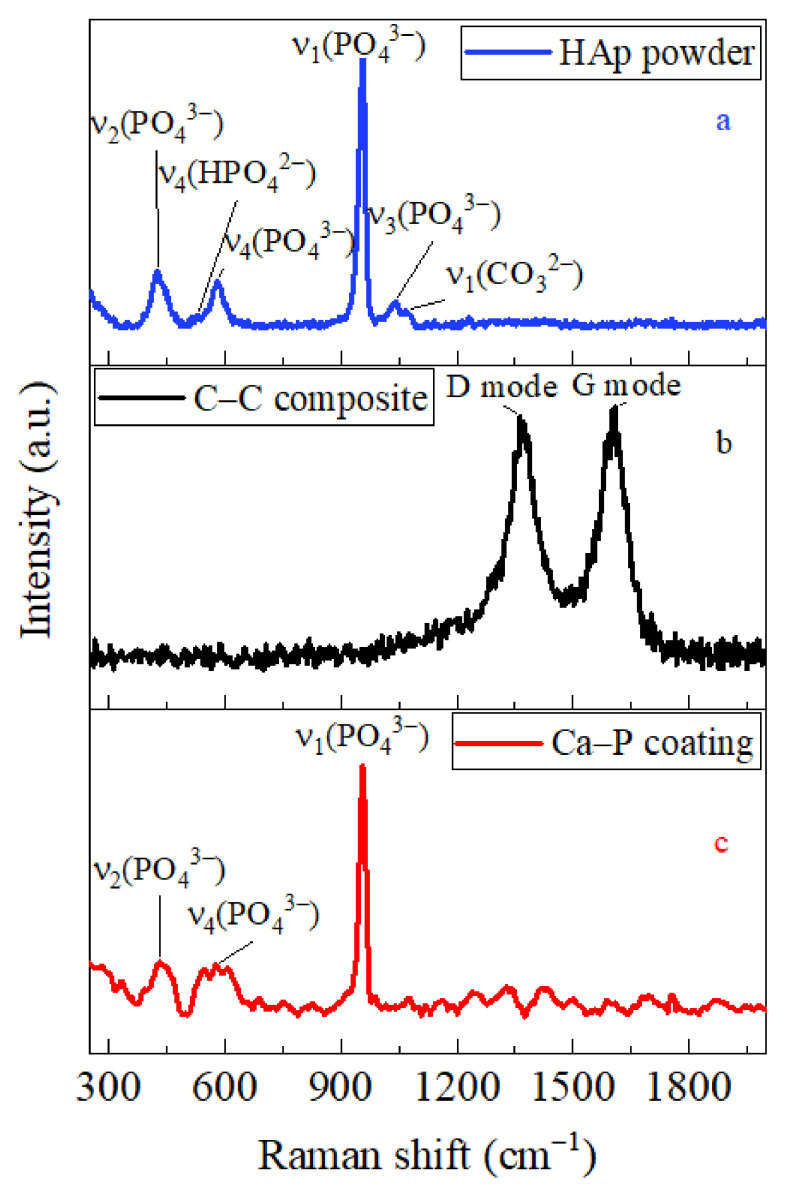
Raman spectra of as-prepared HAp (**a**), C-C composite (**b**) and deposited HAp coating (**c**).

**Figure 3 ijms-25-03375-f003:**
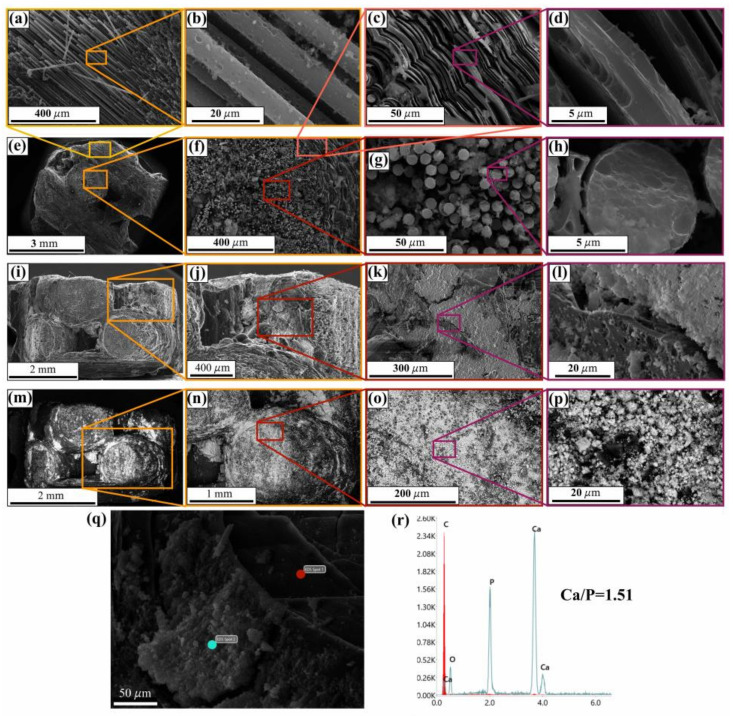
SEM images of HAp-covered composites (**a**–**q**) and EDX spectrum of covered sample (**r**).

**Figure 4 ijms-25-03375-f004:**
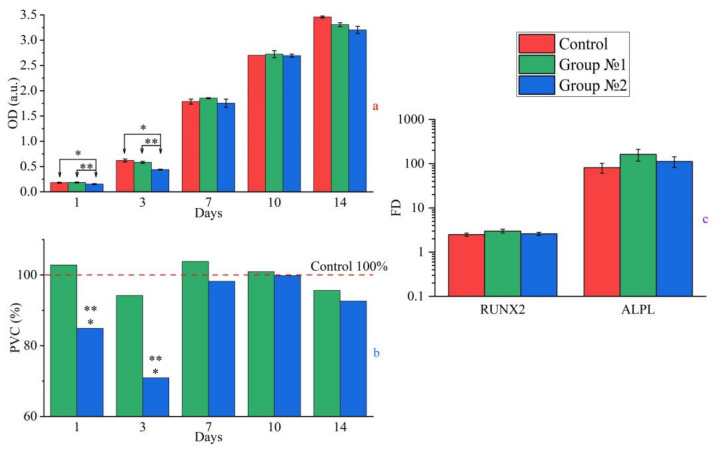
Results of cytocompatibility tests: cell growth diagram (OD values at probe wavelength of 540 nm) (**a**), population of viable cells PVC (**b**) and expression of osteogenic RUNX2 and ALPL genes in progenitor cells of BM MSCs (**c**). Statistically significant results (*n* = 5 per group; *p* * ≤ 0.05) are indicated by one (*; between control and group) and two (**; between group 1 and group 2) asterisks.

**Figure 5 ijms-25-03375-f005:**
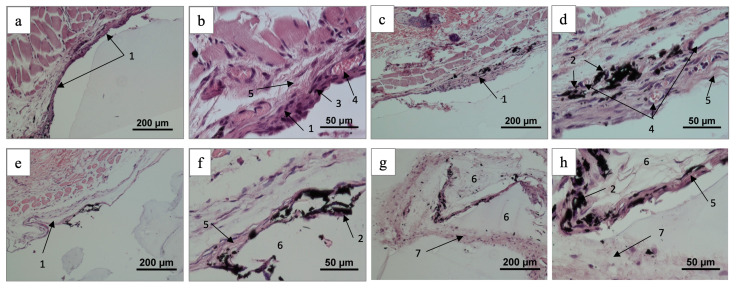
Histological patterns for group 1 post-implantation biopsy (6 weeks—(**a**–**d**); 12 weeks—(**e**–**h**)), hematoxylin–eosin staining: connective tissue capsule (1), dispersed implant material (2), cells of fibroblastic differon (3), capillaries (4), collagen fibers (5), implant macropores (6), and loose connective tissue (7).

**Figure 6 ijms-25-03375-f006:**
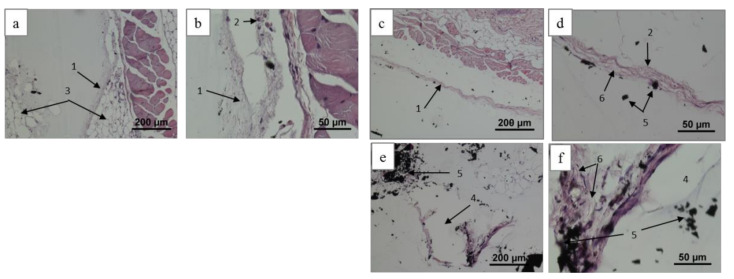
Histological patterns for group 2 post-implantation biopsy (6 weeks—(**a**,**b**); 12 weeks—(**c**–**f**)), hematoxylin–eosin staining: connective tissue capsule (1), capillaries (2), adipose tissue (3), macropores (4), dispersed implant material (5), collagen fibers (6).

**Figure 7 ijms-25-03375-f007:**
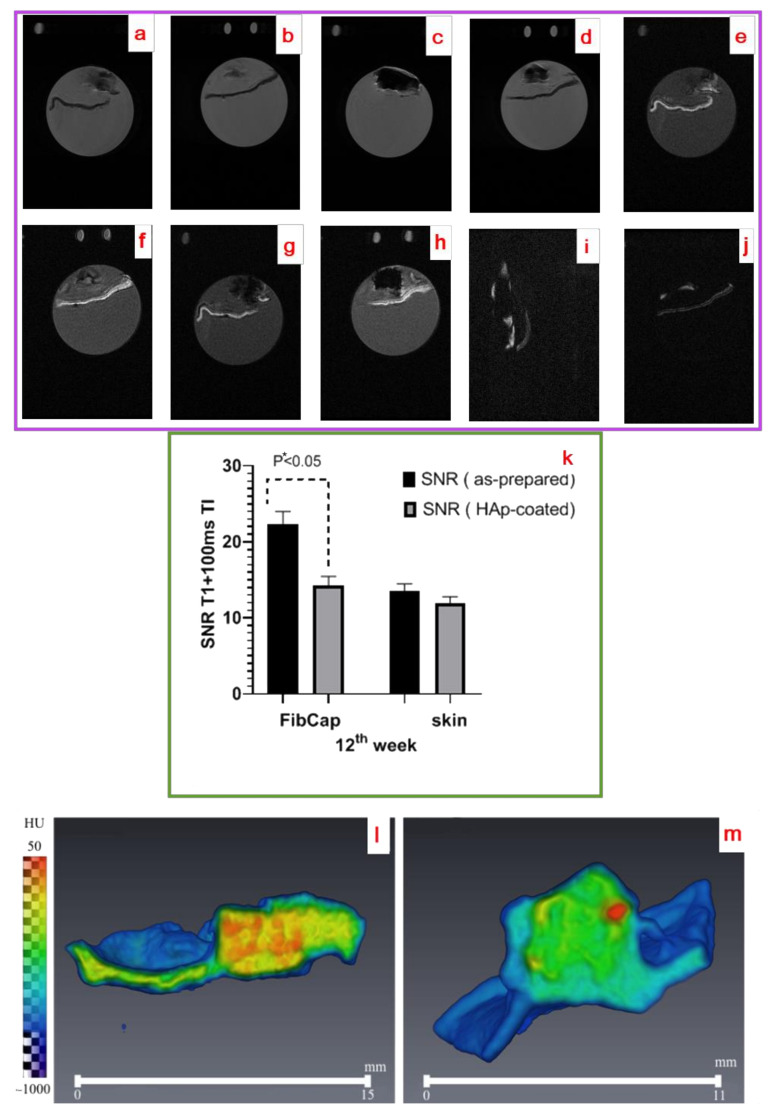
MRI and CT images for the group 1 and 2 post-implantation biopsy: T2w MRI of surface slice + 3 mm from center of group 1 implant (**a**), T2w MRI of surface slice + 2 mm from center implant (group 2) (**b**), T2w MRI of central slice of implant (group 1) (**c**), T2w MRI of central slice of group 2 implant (**d**), T1w MRI of surface slice + 3 mm from center of group 1 implant (**e**), T1w MRI of surface slice + 2 mm from center of group 2 implant (**f**), T1w MRI of central slice of group 1 implant (**g**), T1w MRI of central slice of group 2 implant (**h**), T1w MR of fat suppressed image (TI 100 ms) of group 1 implant (**i**), T1w MR of fat suppressed image (TI 100ms) of group 2 implant (**j**), SNR values (mean ± SD) in fibrosis capsule (FibCap) and skin calculated in ROI (*n* = 10–12) for group 1 (black color) and group 2 (grey color) at 12-th week (**k**), CT pattern around group 1 (**l**) and group 2 (**m**) implants. The difference between the means was statistically significant (one sample *t*-test, *p* * < 0.05).

**Table 1 ijms-25-03375-t001:** Results of the cytotoxicity test: OD, PVC, and TI values. Statistically significant results (*n* = 5 per group; *p* * ≤ 0.05) are indicated by one (*; between control and group) and two (**; between groups) asterisks.

Sample (Group)	24 h			72 h		
	OD, a.u.	PVC, %	TI, %	OD, a.u.	PVC, %	TI, %
CGM (negative control group)	0.252 ± 0.01	100.0	0.0	0.458 ± 0.004	100.0	0.0
As-prepared C-C composite (group 1)	0.245 ± 0.00	97.2	2.8	0.476 ± 0.013	103.9	0.0
HAp-coated C-C composite (group 2)	0.210 ± 0.00 *^,^ **	83.3	16.7	0.389 ± 0.01 *^,^ **	84.9	15.1
50% dimethyl sulfoxide (positive control group)	0.041 ± 0.00 *	16.3	83.7	0.040 ± 0.00 *	8.7	91.3

**Table 2 ijms-25-03375-t002:** Designation of studied gene primers.

Symbol	Encoded Protein	Sequences F and R
RUNX2	Runt- related transcription factor 2 (human)	F: tca-acg-atc-tga-gat-ttg-tgg-g R: ggg-gag-gat-ttg-tga-aga-cgg
ALPL	Alkaline phosphate (human)	F: acc-acc-acg-aga-gtg-aac-ca R: cgt-tgt-ctg-agt-acc-agt-ccc
GAPDH-1	Glyceraldehyde-3-phosphate dehydrogenase, housekeeping gene (human)	F: gaa-ggt-gaa-ggt-cgg-agt-c R: gaa-gat-ggt-gat-ggg-att-tc

## Data Availability

The data presented in this study are available on request from the corresponding author.

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
