# Peer review of "Electrophoretic Deposition of Calcium Phosphates on Carbon–Carbon Composite Implants: Morphology, Phase/Chemical Composition and Biological Reactions"

_ijms, 2024, doi:10.3390/ijms25063375_

Round 1

Reviewer 1 Report

Comments and Suggestions for Authors

Manuscript number ijms-2855128

Title: Electrophoretic Deposition of Calcium Phosphates on Carbon/Carbon Composite Implants: Morphology, Phase/Chemical Composition and Biological Reactions

In this study, the authors are  using the electrophoretic deposition of fine dispersed hydroxyapatite-(HAp) on porous carbon substrates. The resulting specimens were analyzed by SEM, EDS and Raman spectroscopy.  Also in vitro and in vivo tests were performed by the authors.

·         Subsection 2.1. – usually it is used “raw material” or “bulk material” instead of feedstock;

·         Page 3, line 121: Please rephrase “As-have-been-shown-earlier-[28],” with for example “in our previous study [28]”, or “Our previous findings [28] have highlighted…” and so on;

·         Figure 1 – in my opinion the figure caption must be enhanced – EDS analysis is only in Fig1.f . Fig.1.c, Fig.1.d. and Fig.1.e. – each should be defined since some are macroscopic while others are microscopic – please the make the proper changes;

·         Fig.2. – the codification of HAp coating should be used in the entire manuscript. In Fig.2.c. – the authors have used Ca/P coatings …while Ca/P  they are referring to the ration (please see line 121 on page 3). The authors should be consistent with the sample codification ;

·         Page 4, line 133 – please identify on the SEM images the inner circle with an average dimension of ~ 500 μm and the few micron cavities/holes;

·         Page 4, lines 138-140: please verify how the wettability can be confirmed by EDAX analysis;   

·         Page 4 4, lines 142-143: please rephrase. Usually, from the samples manipulation carbon is identified by EDAX. Is it possible that the authors used a salt (Na and Cl) or medium which contain Na and Cl? The authors should  carefully check samples source of contamination. In my humble opinion the authors have identified Na and Cl presence, they can delete the term “atoms”. Regarding the presence of Fe, perhaps the authors should have prepared the surface that they wanted to analysis, in order to eliminate it;

·         In order to confirm the wettability of a surface the authors should perform at least contact angle analysis;

·         Page 4, line 150 – the authors should mention what D and G mode represent;

·         Lines 152-159 – please rephrase: terms such as “caverns”, “whitish HAp” and so on should be replaced with more appropriate ones;

·         How was the Ca/P ratio calculated? (line 168);

·         Or the in vitro tests, I recommend the authors to add discussion on how the HAp coatings has influenced the behavior of the  C/C composite, since this appears to be the aim of the study. It appears that when HAp was added the PVC and TI registered poorer values than the C/C composite, but I couldn’t find the rationality or the cause of this effect.

·         Page 11, Lines 330-331: the authors mention that in comparison to previous studies [23,24], in this study a complex bioactivity and toxicity for the coatings was assessed. I couldn’t find any bioactivity tests; thus, I kindly ask the authors to please verify;

·         Regarding the heat treatment of the HAp coatings, the XRD analysis would have shown the temperature effect along with the type of phases present within the coating. Nonetheless, the authors have also add the word "phase" in the title but without presenting any XRD results.

 Even though interesting, the article should discuss more in depth the obtained results in comparison with other available reports. In its current state form, some parts of the article appear to be more of a report in which the results are presented, rather than discussed and/or correlated with each other or with other relevant studies. This will more likely add value to the manuscript making it more appealing to the readers.

Comments on the Quality of English Language

·         The entire manuscript requires extensive English proofing. Here are some examples:

o   lines 28 – 32 from the abstract sections

o   line 47: ” recipient body”- I assume that the authors are referring to the host body?!?

o   line 56 – “From the one hand…” ,

o   line 64: “On the other hand” – can be eliminated and start with “Another approach for ….”,

o   lines 66-72 must be rephrased, and so on.

·         Overall, the poor English quality makes the manuscript very challenging to follow and to understand.

Author Response

Dear Colleague!

First of all, we would like to thank you for interest in the work.

Concerning you remarks we can note the following.

  • «Subsection 2.1. – usually it is used “raw material” or “bulk material” instead of feedstock;»

Ok. We will use “raw material” phrase.

  • “Page 3, line 121: Please rephrase “As-have-been-shown-earlier-[28],” with for example “in our previous study [28]”, or “Our previous findings [28] have highlighted…” and so on;”

Ok. We will use “in our previous study [28]” phrase.

  • “ Figure 1 – in my opinion the figure caption must be enhanced – EDS analysis is only in Fig1.f . Fig.1.c, Fig.1.d. and Fig.1.e. – each should be defined since some are macroscopic while others are microscopic – please the make the proper changes;”

Ok. We will define and indicate the macroscopic images and microscopic ones. Moreover, we have prepared the composite samples with a diamond-coated tool and ultrasonic cleaning of the samples to avoid contamination. The appropriated SEM image will add.  

  • “Fig.2. – the codification of HAp coating should be used in the entire manuscript. In Fig.2.c. – the authors have used Ca/P coatings …while Ca/P  they are referring to the ration (please see line 121 on page 3). The authors should be consistent with the sample codification “;

Ok. We will note “Ca/P” for the ratio and “Ca-P” for the coatings.

  • “Page 4, line 133 – please identify on the SEM images the inner circle with an average dimension of ~ 500 μm and the few micron cavities/holes;”

Ok. We will indicate them.

  • “Page 4, lines 138-140: please verify how the wettability can be confirmed by EDAX analysis;”

Yes. We are agreeing that EDX does not confirm the wettability. It is an inaccuracy. We will use the term of “adsorption of water vapors from atmosphere”.  

  • Page 4 4, lines 142-143: please rephrase. Usually, from the samples manipulation carbon is identified by EDAX. Is it possible that the authors used a salt (Na and Cl) or medium which contain Na and Cl? The authors should  carefully check samples source of contamination. In my humble opinion the authors have identified Na and Cl presence, they can delete the term “atoms”. Regarding the presence of Fe, perhaps the authors should have prepared the surface that they wanted to analysis, in order to eliminate it;

Yes. It is a very important point. We used the non-metallic diamond-coated cutter and ultrasonic cleaning to avoid contaminations. As a proof, we will add the appropriate SEM and EDX data.

  • “In order to confirm the wettability of a surface the authors should perform at least contact angle analysis;”

As mentioned above, we must discuss the adsorption more than the wettability. EDX are typical for estimation of the adsorption efficiency [https://doi.org/10.1038/s41598-020-74553-4]. Moreover, the measuring of contact angle was difficult with our porous samples due to an easy percolation. That is why there are not the data on a contact angle in the manuscript.

  • “Page 4, line 150 – the authors should mention what D and G mode represent;”

The G mode (≈1600 cm-1) corresponds to the in-plane C-C bond stretching and the D mode (≈1360 cm-1) is caused by the lattice defects [https://doi.org/10.1063/5.0030809]. 

  • “ Lines 152-159 – please rephrase: terms such as “caverns”, “whitish HAp” and so on should be replaced with more appropriate ones;”

Ok. We will use the terms of “gap” and “well-contrasted HAp” instead of the previous ones.

  • “How was the Ca/P ratio calculated? (line 168);”

The Ca/P ratio (in atomic %) was calculated with the EDX data.

  • Or the in vitro tests, I recommend the authors to add discussion on how the HAp coatings has influenced the behavior of the  C/C composite, since this appears to be the aim of the study. It appears that when HAp was added the PVC and TI registered poorer values than the C/C composite, but I couldn’t find the rationality or the cause of this effect.

As found, the PVC and TI values were statistically significant (p<0.05) poorer than the control and the uncovered composite at the earlier stage of the cell growth (24 and 72 hours for cytotoxity and 1 and 3 days for cytocompatibility tests). In the case of the more prolonged cultivation duration (starting from day 7), we detected no significant difference between the control and the groups as well as between group1 and group 2. In our opinion, a probable cause was a presence of a trace amount of technogenic impurities in the samples. The prolonged cultivation with the regular CGM change decreased their concentrations and the PVC values were statistically the same between the groups and the control starting from day 7.      

Ca-P bioceramic coatings could stimulate implant integration into bone tissue but understanding and description of the relevant biological processes are difficult. The presented results could be considered as the necessary preliminary stage of biotesting of the C-C composites covered by HAp. So, as found with the regular MTT, Live/Dead, PC tests, all covered and as-prepared samples were non-toxic (IT≤30 %), cytocompatible (PVC≥70 %) and maintained the osteogenic differentiation of the MSCs. Verification of the osteointegration potential (C-C vs. C-C with HAp) with in vivo models will be in the focus of our following studies.

  • Page 11, Lines 330-331: the authors mention that in comparison to previous studies [23,24], in this study a complex bioactivity and toxicity for the coatings was assessed. I couldn’t find any bioactivity tests; thus, I kindly ask the authors to please verify;

According to our understanding, a bioactive material is a non-toxic material which does not suppress morphogenetic potential of cells (their proliferation and osteogenic differentiation with in vitro tests) and tissues (biocompatibility with in vivo tests).

In the published papers [https://doi.org/10.1016/j.surfcoat.2018.08.052], this term was used for a declaration of bioactivity of the HAp-covered composites with SBF test of biodegradation. As an alternative technique, the MSCs proliferation and alkaline phosphatase activity tests were used earlier for the similar purposes too [https://doi.org/10.1016/j.ceramint.2021.01.032].    

  • Regarding the heat treatment of the HAp coatings, the XRD analysis would have shown the temperature effect along with the type of phases present within the coating. Nonetheless, the authors have also add the word "phase" in the title but without presenting any XRD results.

Despite XRD is a standard of study of phase/thermal transformations, Raman spectroscopy is a reliable tool for characterization for Ca phosphates with determining of tetracalcium phosphates, tricalcium phosphate, HAp, etc. Please, see figure 4 in [10.5405/jmbe.1459] and the interpretation of Figure 9 in [10.1016/j.surfcoat.2011.10.025]. Moreover, Raman spectroscopy is also used for calculation crystallinity index the same as XRD and demonstrates a good efficiency and good resolution compared with XRD [10.1039/c7nj00803a]. That is why we guess the used technique was sufficient for our purposes.  

To confirm Raman data, we will add the XRD pattern in Supplementary.

 Even though interesting, the article should discuss more in depth the obtained results in comparison with other available reports. In its current state form, some parts of the article appear to be more of a report in which the results are presented, rather than discussed and/or correlated with each other or with other relevant studies. This will more likely add value to the manuscript making it more appealing to the readers.

Ok. We will add some discussion and correlation.

The entire manuscript requires extensive English proofing. Here are some examples:

o   lines 28 – 32 from the abstract sections

o   line 47: ” recipient body”- I assume that the authors are referring to the host body?!?

o   line 56 – “From the one hand…” ,

o   line 64: “On the other hand” – can be eliminated and start with “Another approach for ….”,

o   lines 66-72 must be rephrased, and so on.

  • Overall, the poor English quality makes the manuscript very challenging to follow and to understand.

Ok. We will fix occasions. 

Reviewer 2 Report

Comments and Suggestions for Authors

This research group provided a comprehensive study on the electrophoretic deposition of nano-HAp powders on carbon/carbon substrate.  Materials characterization of deposited HAp-C/C composites  were carried out by means of SEM, EDX and Raman. This was then followed by in-vitro tests using MG-63 cells via MTT and PCR assays  as well as in-vivo male BDF1 mice models.  They employed electrophoresis for coating  due to the stability of coated HAp product on carbon substrates when compared to that of plasma-sprayed coating.

The authors had provided a detailed description of the topics in Introduction section, and the experiments were carefully planned.  The results of materials characterization techniques,  in-vitro and in-vivo tests were also well presented and discussed. So it can be published in IJMS after the following minor revision,

(1) In the original submission, only two sentences (lines 362-363)  mentioning electrophoretic deposition of Ca/P layers on carbon implants can stimulate bioresorption in the host body without harmful effects. However, this was not enough for the concluding remark of their investigation. 

The authors should provide a Conclusion section in the revised manuscript describing summaries of their key findings including materials characterization results of the composites, MTT, PCR and histological examination results.

Author Response

“The authors had provided a detailed description of the topics in Introduction section, and the experiments were carefully planned.  The results of materials characterization techniques,  in-vitro and in-vivo tests were also well presented and discussed. So it can be published in IJMS after the following minor revision,

(1) In the original submission, only two sentences (lines 362-363)  mentioning electrophoretic deposition of Ca/P layers on carbon implants can stimulate bioresorption in the host body without harmful effects. However, this was not enough for the concluding remark of their investigation. 

The authors should provide a Conclusion section in the revised manuscript describing summaries of their key findings including materials characterization results of the composites, MTT, PCR and histological examination results.”

Ok. We will expand the Conclusion and Discussion sections.

Round 2

Reviewer 1 Report

Comments and Suggestions for Authors

The manuscript still lacks information and can raise some scientific concerns and, in my opinion, even though the authors have added some information and brought some modifications, the manuscript still must be enhanced.

Thus, I have the following comments for the authors:

Concerning the authors answers to the questions of R1, I must say that the authors haven’t quit managed to offer a point of view for all raised questions and/or comments, and it is was very challenging to identify where new information’s were added within the manuscript.  Usually, the authors should outline each change made (point by point) as raised in the reviewer comments and highlight all the changes made to the manuscript in color so that they are easily seen by the editors/reviewers.

For example, regarding the materials bioactivity I still have some concerns and perhaps the author should better emphasize these aspects, as they understand the definition of “bioactive material” in order to not mislead the audience, namely the difference between bioactivity tests in SBF and the ones carried out in vitro by cell interaction, and in which different types of tests are conducted in order to prove that the material doesn’t induce a negative response when interacting with the host. In my humble opinion, from cell interaction tests it can be assessed  if a material is biocompatible, rather than bioactive. 

Comments on the Quality of English Language

The manuscript still requires major English editing.

Author Response

Please see the attachment. There are the appropriate color indications of remark fixing in the manuscript.

On a separate note, we agree it was untimely to claim about the «bioactivity» without in vivo study of the osseointegration potential into the bone defect of a model animal. We declare the prepared HAp-covered composites as a biocompatible material whose bioactivity will be studied later. We will change the statements.
